# Phytochemical Analysis, In Vitro Anticholinesterase, Antioxidant Activity and In Vivo Nootropic Effect of *Ferula ammoniacum* (*Dorema ammoniacum*) D. Don. in Scopolamine-Induced Memory Impairment in Mice

**DOI:** 10.3390/brainsci11020259

**Published:** 2021-02-19

**Authors:** Nausheen Nazir, Mohammad Nisar, Muhammad Zahoor, Faheem Uddin, Saeed Ullah, Riaz Ullah, Siddique Akber Ansari, Hafiz Majid Mahmood, Ahmed Bari, Abdulrehman Alobaid

**Affiliations:** 1Department of Biochemistry, University of Malakand, Khyber Pakhtunkhwa 18800, Pakistan; mohammadzahoorus@yahoo.com; 2Department of Botany, University of Malakand, Khyber Pakhtunkhwa 18800, Pakistan; mnshaalpk@yahoo.com; 3Department of Engineering, Sarhad University of Information Technology, Peshawar 23000, Pakistan; faheemproudpak@gmail.com; 4Saidu Group of Teaching Hospital Swat, Khyber Pakhtunkhwa 19130, Pakistan; drsaeedullahmerzi@gmail.com; 5Department of Pharmacognosy (MAPPRC), College of Pharmacy, King Saud University, Riyadh 11451, Saudi Arabia; rullah@ksu.edu.sa; 6Department of Pharmaceutical Chemistry, College of Pharmacy, King Saud University, Riyadh 11451, Saudi Arabia; sansari@ksu.edu.sa (S.A.A.); abari@ksu.edu.sa (A.B.); aalobaid1@ksu.edu.sa (A.A.); 7Department of Pharmacology, College of Pharmacy, King Saud University, Riyadh 11451, Saudi Arabia; harshad@ksu.edu.sa

**Keywords:** *Ferula ammoniacum* (D. Don), *Dorema ammoniacum* (D. Don), HPLC, cholinesterases (AChE and BChE), DPPH, ABTS, Y-Maze, NORT, Alzheimer’s disease

## Abstract

Background: *Ferula ammoniacum* (D. Don) is one of the endemic medicinal plants that is traditionally used to treat a number of diseases. Although the plant has been used to enhance memory, the investigational evidence supporting the nootropic effect was unsubstantial. Hence, the rationale for this study was to assess the potential beneficial effect of *F. ammoniacum* seed extracts on learning and memory in mice. Methods: The powdered plant samples (aerial parts) were subjected to extraction ad fractionation. Among the extracts, crude and ethyl acetate extracts were screened for major phytochemicals through HPLC analysis. All the extracts were evaluated for the in vitro anticholinesterase (AChE and BChE) and antioxidant potentials. Among the extracts the active fraction was further assessed for improving learning and memory in mice using behavioural tests like Y-maze and novel object recognition test (NORT) using standard protocols. After behavioural tests, all the animals were sacrificed and brains tissues were assessed for the ex vivo anticholinesterase and antioxidant potentials. Results: Phytochemicals like chlorogenic acid, quercetin, mandelic acid, phloroglucinol, hydroxy benzoic acid, malic acid, epigallocatechin gallate, ellagic acid, rutin, and pyrogallol were identified in crude methanolic extract (Fa.Met) and ethyl acetate fraction (Fa.EtAc) through HPLC. Fa.EtAc and Fa.Chf extracts more potently inhibited AChE and BChE with IC50 values of 40 and 43 µg/mL, and 41 and 42 µg/mL, respectively. Similarly highest free radical scavenging potential was exhibited by Fa.EtAc fraction against DPPH (IC50 = 100 µg/mL) and ABTS (IC50 = 120 µg/mL). The extract doses, 100 and 200 mg/kg body weight significantly (*p* < 0.01) improved the short-term memory by increasing the percent spontaneous alternation in the Y-maze test along with increasing discrimination index in the NORT that clearly indicated the enhancement in the recognition memory of mice. Conclusion: The extracts more potently scavenged the tested free radicals, exhibited anticholinesterase activities, improved the learning abilities and reduced the memory impairment induced by scopolamine in mice model thus suggesting that these extracts could be effectively used for the management of oxidative stress, neurodegenerative diseases and memory loss.

## 1. Introduction

According to a 2001 World Health Organization (WHO) report, 25% of the world population is suffering from neurological disorders like epilepsy, headache, migraine, madness, insomnia, stress, depression, Alzheimer’s (AD), and Parkinson’s diseases [1]. AD is a chronic neurodegenerative disease where the deposition of neuronal amyloid along with a deficit of neurotransmitters; acetylcholine (ACh) level occurs, leading to memory dysfunction [2]. Oxidative stress has been pointed to be main factor involved in the pathogenesis of AD. Another predominant reason is the reduction of acetylcholine level in the brain, which is the most notable biochemical change in AD. Acetyl cholinesterase (AChE) and butyryl cholinesterase (BChE) are the key enzymes that causes the degradation of ACh resulting in cholinergic deficits and termination of cholinergic neurotransmission. Thus, the inhibition of cholinesterases is considered to be the base while treating AD and this promising strategy is clinically applied everywhere for treating neurodegenerative diseases and is a rational approach. Inhibition of brain cholinesterases (AChE and BChE) by potential inhibitors reinstates the level of ACh and, thus, plays an important role in the treatment of AD, Dementia, and Parkinson’s disease [3] and the consequences of cholinesterases inhibition are assessed by observing cholinergically mediated processes such as cognition [4]. Acetylcholine is a key neurotransmitter in the nervous system that interacts with receptors associated with processes of learning and memory. It has been proved that cholinesterase inhibition is a preferable direct receptor agonist therapy that relives ACh deficit and stimulates either nicotinic or muscarinic ACh receptors. Cholinesterases inhibitors like Donepezil are approved by the Food and Drug Administration for the symptomatic treatment of AD. These drugs improve cognitive and neuropsychiatric symptoms. Donepezil is a second-generation cholinesterases inhibitor with a high selectivity in the central nervous system, can reversibly inhibit cholinesterases (AChE and BChE) whereby reducing ACh degradation, thus increase neurotransmitter concentration in the synaptic cleft that prolongs its duration of action, ultimately elevating the central cholinergic activity to improve cognitive function. The use of donepezil seems to have a positive effect on the functioning and enhancements in cognition [5].

Researchers these days are in search of plants-based cholinesterase inhibitors and antioxidants which would significantly recovers cognitive decline. Although synthetic antioxidants are utilized in the food industry, but they have exhibited adverse side effects including liver damage and carcinogenesis. Natural sources, especially plants, provide a miscellaneous and largely unexploited reservoir of bioactive phytochemicals for drug discovery and for the development of new cholinesterase inhibitors and antioxidants. Previous studies have highlighted the potential beneficial effects of plants as vital sources for cholinesterase inhibitors [6]. Individuals suffering from neurological disorders in the developing and underdeveloped countries primarily rely on traditional herbal medicines [7]. The folkloric knowledge about plants have helped in isolation of a large number of bioactive compounds from various medicinal plants for example Galantamine, a potential inhibitor of acetylcholinesterase that have been isolated from *Galanthus nivalis* and is frequently used to treat AD [8]. 

The association of neurodegenerative diseases like AD with some pathological conditions, such as impaired mitochondrial function, amyloids–beta (αβ) deposition, neuroinflammation, cholinergic deficit, and oxidative stress has been pointed out by many researchers [9,10]. In patients with AD, intense AChE, BChE and free radical activities have been found in the area of amyloid plaques and neurofibrillary tangles. Amyloids-beta (αβ) and cholinesterase activities may act as a physiological modulator of cholinergic function and induce neurotoxicity, alongside inhibiting the synthesis and release of ACh, resulting in cholinergic hypofunction, reduced neural efficiency, and cognitive impairment. These effects seem to appear in the cortex and hippocampus regions but not in other brain areas. αβ activates microglial cells to act as pro-inflammatory cells that secrete pro-inflammatory cytokines, which induces further αβ deposition. Microglia cells are the resident immune cells of the central nervous system which express surface receptors that activate or amplify the innate immune response. During cellular damage microglial cells respond quickly by inducing a protective immune response, which result in the up regulation of inflammatory molecules as well as neurotrophic factors. However, in case of chronic inflammation the prolonged activation of microglia leads to the production of a wide array of neurotoxicins, proinflammatory cytokines such as interleukin-1 (IL-1β), interleukin-6 (IL-6), tumour necrosis factor alpha (TNFα), and cholinesterases (AChE and BChE) in the extracellular spaces [9,10]. It has been proposed that the increase microglial cells activation in brain may be one of the early events that leads to oxidative damage and is considered to be the most abundant source of free radicals in the brain (such as superoxide and nitric oxide). Microglial cell derived radicals, as well as their reaction products such as hydrogen peroxide and peroxynitrite have been shown to be involved in oxidative damage and neuronal cell death in neurological diseases such as AD. Studies have shown that microglia cells play a role in supporting cognitive processes and homeostasis in the healthy adult brain while their absence results in cognitive and learning deficits in rodents during development. It also should be noted that microglial cells have efficient antioxidative defence mechanisms but when the production of reactive oxygen species (ROS) is prolonged, the endogenous reserves of antioxidants become exhausted and result in cellular damage [9].

ROS are constantly generated in redox processes during metabolism that attacks the biomolecules like enzymes, lipids, proteins, DNA and RNA resulting in an irreversible damage to these important commodities of life. Luckily, the human body can manage to cope with oxidative stress, through a number of defensive systems involving oxidant enzymes, and non-enzymatic compounds. However, in pathological conditions where excessive amounts of ROS encounters the defensive systems cannot properly manage the situations leading to chronic diseases like cancer, atherosclerosis, nephritis, diabetes mellitus, rheumatism, ischemic, cardiovascular disease, and aging. Oxidative stress also plays a key role in the development of neurodegenerative disorders like AD and Parkinson’s diseases [11]. The accumulation of ROS has been found in several chronic diseases, including AD suggesting that ROS may contribute to the pathogenesis of these diseases by inducing oxidative stress. Medicinal plants counteract the potential damaging effects of oxidative stress by the production of antioxidants. Previous studies have revealed that antioxidants have significant potential to reduce the symptoms and prevalence of AD [12]. 

Antioxidant like phenolics and flavonoids are commonly found in various fruits and vegetables and they have been shown to provide a fruitful defence against oxidative stress from oxidizing agents/free radicals. Oxidative stress is often defined as an imbalance between free radicals and antioxidant defence system. These radicals contribute to the development of many diseases including AD and can be neutralized by antioxidants. The consumption of fruits and vegetables are beneficial for health as they contain various secondary metabolites and other nutrients capable of curing a number of health complications [13]. Due to toxic effects of synthetic antioxidants; recent studies are mainly focused on their replacement with naturally occurring antioxidants from medicinal plants. Therefore, it is important to discover and isolate novel bioactive compounds from natural sources and evaluate their various biological potentials. Secondary metabolites like flavonoid and phenolics have great importance as antioxidants as they have high potentials of scavenging free radicals like ROS [14].

*Ferula ammoniacum* (D. Don) *Spalik*, *M. panahi*, *Piwczynski* and *Puchalka*, is an important members of family Apiaceae also known as *Dorema ammoniacum* (D. Don) [15], has a wide variety of traditional uses, for example, it produces gum which exude from stem, roots, and petioles that is medicinally used to treat a number of diseases. This gum resin is traditionally used as a carminative, diaphoretic, mild diuretic, expectorant, stimulant, cardiovascular, anthelmintic, antiepileptic, asthmatic, and antitussive. *D. ammoniacum* gum resin has anti-helminthic potentials and has been used to treat gastrointestinal disorders and as anti-inflammatory agent in Iranian traditional medicine [16]. Its latex has also medicinal uses like expectorant, bronchitis, and stomach-ache [17]. Ghasemi et al. (2018) elucidates the electrophysiological mechanism of the effect of *D. ammoniacum* gum on a cellular model of epilepsy, using intracellular recording method [18]. *D. ammoniacum* gum resin has shown a significant anticonvulsant activity in pentylentetrazole model (in mice) in a dose dependent manner via involvement of GABAergic and opioid systems [19]. The gum resin has exhibited significant acetylcholine esterase inhibitory potential [20] and antimicrobial activities [21]. Apart from the mentioned potentials, *D. ammoniacum* have also exhibited other pharmacological effects, including anxiolytic, anti-inflammatory, anti-coagulation, anti-epileptogenic and anticonvulsant [17]. In Persian traditional medicine textbooks, *D. ammoniacum* is one of the materia medica that is frequently used for stroke and paralysis as it opens the blood vessels and is also considered as a thrombolytic agent. In traditional medicines, it is used as a remedy by itself and sometimes it is used in combination with other drugs to treat health complications like stroke and paralysis [22]. The essential oil of *D. ammoniacum* leaf, fruit, and stem have already been reported to have antioxidant, antimicrobial, and low cytotoxicity activities [23,24]. Chemical composition of essential oil and toxicological studies like acute and sub-acute toxicity of *D. ammoniacum* oleo-gum-resin has also been reported [25]. Antibacterial and vasodilatory effects of this herbal plant have also been reported [26]. *D. ammoniacum* aerial part extract has been used as a mediator in the synthesis of silver nanoparticles that have displayed antimicrobial activities [27]. The *Dorema* other species are also traditionally used as anticancer, antimicrobial, insecticidal, and as antioxidant agents [28,29,30]. However, its effect as nootropic agent has not been investigated yet. 

Therefore, the current investigational study was designed to evaluate the phytochemical composition and evaluate in in vitro its anticholinesterase and antioxidant potential along with in vivo nootropic activities in mice. 

## 2. Material and Methods

### 2.1. Drugs and Chemicals

AChE (*Electron eel* type-VI-S) and Aquine BChE along with their substrates acetylthiocholine iodide and butyrylthiocholine iodide, and potassium phosphate buffer (pH 8.0) were purchased from Sigma Aldrich St. Louis, MO, USA. Antioxidant chemicals: DPPH, ABTS, ascorbic acid, malic acid, morin, epigallocatechin gallate, pyrogallol, rutin, quercetin, chlorogenic acid, mandelic acid, hydroxy benzoic acid, and gallic acid along with Folin–Ciocalteu regent were acquired from Sigma Aldrich, Darmstadt, Germany. Donepezil and scopolamine were also purchased from Sigma Aldrich (USA). All these chemicals used were of analytical grade; however, the HPLC solvents were of HPLC grade. 

### 2.2. Plant Material Collection and Identification

*Ferula ammoniacum* aerial parts were collected from the hilly areas of village Yar Khan Banda (Kondoly), Timergara Dir (Lower), Pakistan in 2019. The plant was identified by plant taxonomist Prof. Mehboob-ur-Rahman, PGC, Swat, Khyber Pakhtunkhwa, Pakistan. The plant specimens were deposited with voucher number BGH.UOM.163 in the Botanical Garden Herbarium, University of Malakand, Pakistan. 

### 2.3. Preparation of F. ammoniacum Aerial Parts Extract/Fractions 

The collected samples were cleaned and kept on a clean paper to shade dried for 20 days. The dried aerial parts sample were crushed through mechanical grinder. Approximately 3 kg of powder sample were subjected to maceration in 80% methanol for 14 days with periodical shaking [13]. Filtration was conducted through muslin cloth followed by filtration through Whattman filter paper. The filtrate was concentrated into a semisolid mass using rotary evaporator (Heidolph Laborota 4000, Schwabach, Germany) under reduced pressure and then completely dried through lyophiliser. The semisolid mass was then solidified (280 g end yield product) in open air. The crude methanolic extract (Fa.Met) 250 g was subjected to fractionation using solvent-solvent extraction procedure starting from a low polarity solvent to highly polar one. The final yield of *n*-hexane (Fa.Hex), chloroform (Fa.Chf), ethyl acetate (Fa.EtAc), *n*-butanol (Fa.Bn), and aqueous (Fa.Aq) fraction were 35, 22, 45, 22 and 113 g, respectively. 

### 2.4. HPLC-UV Characterization of Phytochemicals

To prepare the sample for HPLC analysis, about 1 g of extract was mixed with 20 mL of methanol and water (1:1 *v*/*v*) and heated on a water bath at 70 °C for 1 h. Centrifugation of sample was conducted at 4000 rpm for 10 min after cooling. Then, 2 mL from sample was filtered with Whattman filter paper into HPLC vials and labelled with proper codes.

An Agilent 1260 infinity High-performance liquid chromatography (HPLC) system was used to analyse the samples while separation was achieved via Agilent Zorbax Eclipse XDB-C18 column with gradients system comprising of solvent A (methanol:acetic acid:deionized water, 100:20:180, *v*/*v*) and solvent B (methanol:acetic acid:deionized water, 900:20:80, *v*/*v*) used to elute the bioactive compounds [31]. The concentration of a given phytochemicals was determined using following single point calibration formula: (1)$$Cx=\frac{Ax\times Cs\left(\mu g/\mathrm{mL}\right)\times V\left(\mathrm{mL}\right)}{As\times Sample\;\left(\mathrm{wt}.\;\mathrm{in}\;g\right)}$$
where: *Cx* = Sample concentration; *As* = Standard peak area; *Ax* = Sample peak area; *Cs* = Standard concentration (0.09 µg/mL).

### 2.5. Assessment of Total Phenolic Contents 

Total phenolic content (TPC) in the extracts was determined using previously reported method [32]. In this test, extract/fractions samples (100 µL), distilled water (500 µL), Folin–Ciocalteu reagent (100 µL), and 7% sodium carbonate (1000 µL) were mixed together and allowed to stand for 90 min. Finally, absorbance was measured at 760 nm using UV-Spectrophotometer. Gallic acid standard curve was obtained using the dilutions: 1000, 500, 250, 125, 62.5, and 31.05 µg/mL. The TPC was expressed as mg of gallic acid equivalent per gram (mg GAE/g) of dry sample (averaged from three parallel measurements). 

### 2.6. Assessment of Total Flavonoid Contents 

Total flavonoids contents (TFC) in *F. ammoniacum* aerial parts extract/fractions were calculated using previously reported method [32]. Quercetin was used as a standard and TFC was determined as mg of Quercetin equivalent (mg QE/g) per gram of dry sample of extract/fractions. A calibration curve for Quercetin was obtained using their various dilutions (1000, 500, 250, 125, 62.5, and 31.05 µg/mL). about 100 µL from each sample dilutions were taken and mixed with distilled water (500 µL), 5% sodium nitrate (100 µL), 10% aluminum chloride (150 µL) solution, and 1 M sodium hydroxide (200 µL), then allowed to stand for 5 min and its absorbance was recorded at 510 nm using UV-Spectrophotometer. All results were taken in triplicate.

### 2.7. In Vitro Cholinesterase Inhibition Potential of Extracts

Ellman assay [33] was used to assess the *F. ammoniacum* aerial parts methanolic extract/fractions for their cholinesterase inhibition potentials. About 205 µL extract/fractions and 5 µL of AChE (0.03 U/mL)/BChE (0.01 U/mL) along with 5 µL DTNB were taken in a cuvette and incubated at 30 °C for 15 min. After incubation, 5 µL acetylthiocholine iodide or butyrylthiocholine iodide (substrate) were added to the mixture that resulted in yellow coloration due to formation of 5-Thio-2-nitro benzoate anion. Then, the absorbance of the resulting mixture was measured at 412 nm through double beam spectrophotometer (Thermo electron-corporation, Waltham, MA, USA). As a negative control, a solution containing all the above-mentioned components except plant extracts/fractions were mixed together. The same procedure mentioned above was used to constitute the reaction mixture of positive control galantamine and absorbance was measured at 412 nm. For each sample, absorbance was recorded for 4 min. Percent enzyme activity and inhibition potential of both enzymes were measured using the following formulae:(2)V=Δ Abs/Δt
(3)$$\%\;\mathrm{Enzyme}\;\mathrm{activity}=\frac{V}{Vmax}\times 100$$
(4)% Enzyme inhibition=100−% enzyme activity
where: *V* shows the rate of reaction in the presence of inhibitor and *Vmax*, the rate of reaction in its absence.

### 2.8. DPPH (2,2-Diphenyl-1-picrylhydrazyl) Free Radical Scavenging Potential of Extracts

Brand-Williams assay [34] was used to determine the free radical scavenging potential of *F. ammoniacum* aerial parts methanolic extract/fractions against DPPH. The synthetic free radical, DPPH solution was prepared by dissolving 24 mg of it in 100 mL of methanol. The plant samples (1 mg/mL) were also made in methanol and working solutions were prepared using serial dilutions in the concentration range of 31.05–1000 µg/mL. About 0.1 mL of each working dilution was mixed with 3 mL of DPPH solution and incubated for 30 min at 25 °C. Absorbance was measured at 517 nm using UV-spectrophotometer (Thermo Electron Corporation, Beverly, MA, USA). Ascorbic acid was used as a positive control. All results were taken in triplicates and are presented as mean ± SEM. Percent radical scavenging activity was calculated using the following equation:(5)$$\%\;\mathrm{Free}\;\mathrm{radical}\;\mathrm{scavenging}\;\mathrm{potential}\;=\frac{\mathrm{Blank}\;\mathrm{sample}\;\mathrm{absorbance}-\mathrm{sample}\;\mathrm{absorbance}}{\mathrm{Blank}\;\mathrm{sample}\;\mathrm{absorbance}}\times 100$$

### 2.9. ABTS (2,2′-Azinobis-3-ethylbenzothiazoline-6-sulfonic Acid) Free Radical Scavenging Potential of Extracts

According to the previously reported Re et al. [35] method antioxidant activity of the selected plant methanolic extract/fractions were determined against ABTS free radical. ABTS (7 mM) and Potassium per sulphate (2.45 mM) solutions were prepared and mixed thoroughly. For the production of ABTS free radicals in the solution, the mixture was kept in dark for overnight at room temperature. After incubation absorption of a 3 mL volume was adjusted 0.7 (noted at 745 nm) by adding methanol. To determine the free radical scavenging ability about 300 µL of extract/fractions working dilutions and 3 mL of ABTS solution were mixed thoroughly and incubated for about 6 min. Finally, absorbance of the resulting mixtures was recorded at 745 nm using a double-beam spectrophotometer. Ascorbic acid was used as a positive control. Results were taken in triplicates and % ABTS scavenging potential was calculated using Equation (5). 

### 2.10. In Vivo Studies

#### 2.10.1. Experimental Animals

The animals used in the in vivo study were Swiss male albino mice (26–31 g body weight) that were obtained from National Institute of Health Islamabad, Pakistan. All the animals were maintained in the animal house of the Department of Pharmacy, University of Malakand. All the mice were divided into six groups and housed in individual cages. All the animals were kept under normal laboratory conditions of light/dark cycle (12/12 h) and were provided water and normal pellet diet. All the animal procedures were conducted according ARRIVE guidelines and also the approval was taken from the Departmental Animal Ethical Committee (DAEC/2019/1), University of Malakand, Pakistan. 

#### 2.10.2. Acute Toxicity Studies of the Fa.EtAc Fraction 

The acute oral toxicity study on the plant extract (Fa.EtAc) was investigated according to the Organization for Economic Co-operation and Development (OECD) guideline 423. All animals (*n* = 6) were treated orally with a single dose of 2000 mg/kg body weight of Fa.EtAc (the most active fraction) to 3 animals, and the other 3 animals receiving distilled water at 10 mL/kg to evaluate the toxic effects in any in the experimental animals. Immediately after dosing, the mice were observed continuously for 2 h for any symptoms of toxicity such as convulsions, loss of righting reflex, motor activity, muscle spasm, tremors, lacrimation, sedation, hypnosis, diarrhoea, and salivation. Mice were then kept under observation up to 14 days for any signs of toxicity or mortality. The Fa.EtAc remained safe and nontoxic up to the dose range of 2000 mg/kg body weight. The toxicity assessment was accomplished following guidelines of the Organization for Economic Cooperation and Development (OECD). The animals were administered Fa.EtAc 200 mg/kg body weight as 1/10th of the highest dose for multiple dose investigation to assess the in vivo neuroprotective and nootropic activity [36]. 

#### 2.10.3. Experimental Design 

After the acclimatization period, in the scopolamine-induced amnesia test the Albino male mice were randomly divided into six groups (*n* = 6), of eight animals each for administration of Fa.EtAc fraction and were given doses as per following details: 

Group I was used as normal control group and were given normal saline (8 mL/kg, p.o.) solution. 

Groups II (negative control) received scopolamine 1 mg/kg body weight intraperitoneally.

Group III was used as positive control, and were given donepezil (2 mg/kg, p.o) and scopolamine (1 mg/kg, i.p.). 

Group IV, V, and VI were used as treatment groups and were treated with Fa.EtAc fraction with different doses. Group IV were treated with 50 mg/kg body weight (b.w) Fa.EtAc (p.o) and scopolamine (1 mg/kg, i.p.). Group V were treated with 100 mg/kg Fa.EtAc (p.o) and scopolamine (1 mg/kg, i.p.), while Group VI were given with 200 mg/kg Fa.EtAc (p.o) and scopolamine (1 mg/kg, i.p.).

The dosing details are also given in Table 1. The volume of oral (p.o.) and intraperitoneal (i.p.) administrations was 1 mL/100 g b.w of mice. The treatments were continued for 8 days. In the behavioural Y-maze and Novel object recognition tests, scopolamine (1 mg/kg, i.p.) was given to these groups only on day 8th, and according to the standard procedure, 1 h after the administration of drugs all the animals were subjected to Y-maze and Novel object recognition test. 

### 2.11. Behavioural Assessment

#### 2.11.1. Y-Maze Spontaneous Alternation Behaviour Test

Y-Maze spontaneous alternation behaviour test was used to evaluate the short-term memory potentials (STM) of mice by recording spontaneous alternation in a single session on day 10 [36]. The Y-maze apparatus is comprised of three arms of equivalent size, labelled as A, B, and C, respectively. Each arm is 20 cm long, 6 cm wide and 15.5 cm high and is oriented at an angle of 120° from the other two. One hour after the last treatment and 30 min after scopolamine injection (except for the distilled water group), all the mice were allowed to move freely through the maze for 8 min. The numbers of arm entries were recorded for each mouse. An arm entry was considered to be completed when the hind paws of the mouse were completely placed in an arm. Spontaneous alternation was defined as consecutive entries in a specific sequence of arm transitions (ABC, BCA, or CAB but not BAB or CAC or CBC) by a mouse into the three different arms that reflects short-term memory. The total number of arm entries reveals an overall locomotor activity. The arms of the maze were cleaned using 70% *v*/*v* ethanol between trials to avoid olfactory cues. The number of arm entries, same arm returns (SAR), and alternate arm returns (AAR) were measured. The percentage of spontaneous alternation performance (% SAP) was determined using the following equation.
(6)%SAP= [(Number of alternations)/(Total arm entries − 2)] × 100

#### 2.11.2. The Novel Object Recognition Test and Novel Object Location Tests

The novel object recognition test (NORT) was conducted in order to evaluate recognition memory in mice [36]. The apparatus was comprised of a white coloured plywood box (40 cm × 40 cm × 66 cm) with a network floor that was carefully cleaned with 70% *v*/*v* ethanol after every trial. The apparatus was illuminated with a 60 W light suspended 50 cm over the crate. The arena of the apparatus and objects were cleaned with 70% *v*/*v* ethanol between trials to avoid olfactory cues. NORT consisted of habituation, sample, and test phases. For behavioural assessment, 1 h after dosing the mice were habituated to the experimental apparatus in the absence of objects twice a day (with 5 h interval) in a 10 min session for three consecutive days to explore the objects. On the fourth day, two similar objects were placed in two opposite places of the box and exploration was recorded for 15 min session called sample phase (first trial: T1). After this session the animals were kept for a retention break of 1 h. Exploration was considered when an animal touches the object or it directs its nose at a distance less than 2 cm to the object. The mouse was then returned to its home cage after test. At the 2nd day of the test (day 10th of drug treatment), 30 min after scopolamine injection (except for the distilled water group) called test phase (second trial: T2), each mouse was placed again in the open field in which a novel object (plastic square) was replaced by one of the objects placed in the 1st day trial and mice were left individually in the box for 5 min. The location of the object was counterbalanced so that one half of the mice in each group saw the novel object on the left side of the box arena, and the other half saw the novel object on the right side of the box arena to eliminate bias of sides. The time spent by the animal for exploring the novel object (N) and the familiar (F) objects was recorded during 5 min. A mouse was scored as exploring when its head was oriented towards the object within a distance of 2 cm or when the nose was in contact with the object. Parameters including the time (seconds) spend in exploring familiar (F) object, time (in seconds) spend on exploring the novel (N) object, and total time (in seconds) spend on exploring both objects (N + F) were measured separately. Percentage of discrimination index (DI) was calculated by the following equation:(7)$$\%\mathrm{DI}=\frac{\left(N-\;F\right)}{\left(N+\;F\right)}\times 100$$

The same apparatus was used for the assessment of novel object location task in experimental animals. In the sample phase, animals were exposed to two similar objects in the same location (initial location) for 15 min. After 1 h retention phase, one object is changed to a novel position while the other was kept on its initial position. Exploration time for assessing the novel location (NL) and familiar location (FL) was determined for 15 min in the test phase [37]. Percent exploration time was determined by the following formula: (8)$$\mathrm{Exploration}\;\mathrm{time}\;\%=\;\frac{\left(\mathrm{NL}\right)}{\left(\mathrm{NL}+\;\mathrm{FL}\right)\;}\times 100$$

#### 2.11.3. Isolation of Frontal Cortex and Hippocampus

Immediately after the Y-Maze test and NORT, all the animals were sacrificed by cervical dislocation before decapitation to provide each animal with a quick and painless death according to the procedure illustrated in schedule-1 of UK (animal scientific procedure act 1986) and the brain of each animal was carefully isolated. The frontal cortex (FC) and hippocampus (HC) were dissected out in ice cold phosphate buffer saline (0.1 M; pH 8.0). The tissues were weighed and 20 mg tissue/mL homogenate of brain samples was prepared by homogenizing the isolated parts in phosphate buffer (pH 8.0). The homogenates were centrifuged at 10,000 rpm for 10 min at 4 °C, and finally supernatants were collected at a standardized protein content of 5 mg/mL that was used for the estimation of ex vivo cholinesterase activity following Elman’s assay [33] and antioxidants activity using Brand William’s assay [34]. 

### 2.12. Statistical Analysis

All the in vitro and in vivo experiments were performed in three replicate and results are presented as Mean ± SEM. The Student’s t-test and one-way ANOVA followed by Dunnett’s post hoc multiple comparison test have been used. *p* ≤ 0.05 were considered as significant. Linear regression was used to calculate IC_50_ for % DPPH, ABTS, AChE, and BChE inhibition against the different concentration of test samples by means of Excel program 2007. 

## 3. Results

### 3.1. Identification and Quantification of Phytochemicals Compounds

A typical HPL-UV chromatogram of *F. ammoniacum* aerial parts methanolic and ethyl acetate extracts have been shown in Figure 1. A total of five phytochemicals were identified in methanolic extract while in ethyl acetate fraction. The detailed identification of each antioxidant with their respective peak position in chromatogram and retention time (Rt) is given in Table 2. Chlorogenic acid, quercetin, mandelic acid, phloroglucinol, and hydroxy benzoic acid eluted at retention times of 6.5, 10.5, 30.5, 35.5, and 36.3 min were present in Fa.Met, while malic acid, chlorogenic acid, epigallocatechin gallate, quercetin, ellagic acid, rutin, pyrogallol were identified in Fa.EtAc fraction at retention times of 3.1, 6.5, 8.0, 10.5, 16.6, 22.7 and 28.1 min. These compounds are the possible compounds as on same retention time different compounds can come out from column even under the same HPLC method. The concentrations of possible phytochemicals in crude extracts were: 9.203, 0.008, 0.014, 14.953, and 93.825 µg/mL (chlorogenic acid, quercetin, mandelic acid, phloroglucinol, and hydroxy benzoic acid, respectively) as presented in Table 2. 

### 3.2. Total Phenolic Content

Results of TPC in the crude methanolic extract and various fractions of *F. ammoniacum* aerial parts are presented in Table 3. A standard gallic acid curve was constructed by preparing the dilutions 20, 40, 60, 80 and 100 mg/mL to estimate the TPC using regression equation. The phenolic contents of Fa.Met, Fa.Hex, Fa.Chf, Fa.EtAc, Fa.Bn and Fa.Aq were 68.25 ± 1.14, 65.55 ± 0.84, 75.12 ± 1.58, 88.23 ± 1.13, 55.18 ± 0.68, and 51.75 ± 1.87 mg GAE/g of dry sample, respectively. Out of them Fa.EtAc, Fa.Bn, and Fa.Chf fractions were rich in total phenolic contents.

### 3.3. Total Flavonoid Contents

To estimate the TFC in *F. ammoniacum* aerial parts, the quercetin calibration standard curve was used for which dilutions 20, 40, 60, 80 and 100 mg/mL were prepared. TFC were: Fa.Met, 66.97 ± 0.99, 68.01 ± 1.17, 77.83 ± 2.16, 85.93 ± 0.67, 45.51 ± 1.99 and 59.79 ± 1.39, respectively, in Fa.Hex, Fa.Chf, Fa.EtAc, Fa.Bn and Fa.Aq (Table 3). Highest total flavonoid content were calculated in Fa.EtAc, Fa.Bn, and Fa.Chf fractions. 

### 3.4. In Vitro Cholinesterase Inhibitory Potential of F. ammoniacum Aerial Parts Methanolic Extract/Fractions 

Cholinesterase (AChE and BChE) inhibitions of *F. ammoniacum* methanolic extract/fractions were determined at various concentrations and are presented in Figure 2A,B. Among different fractions of *F. ammoniacum*, Fa.EtAc and Fa.Chf showed prominent inhibition against AChE (89.22 ± 1.74 and 88.44 ± 0.80) at highest concentration (1000 µg/mL) with IC_50_ values 40 and 43 µg/mL, respectively. Galantamine was used as a standard and showed percent inhibition of 96.56 ± 1.08 against AChE (IC_50_ = 30 µg/mL) at highest concentration 1000 µg/mL (Figure 2A). Similarly, Fa.EtAc and Fa.Chf fractions also showed highest inhibition against, BChE which were 86.37 ± 0.61, and 85.31 ± 0.49 at highest concentration of 1000 µg/mL with IC_50_ values of 41 and 42 µg/mL, respectively (Appendix A). The positive control galantamine showed percent inhibition of 95.17 ± 071 against BChE at 1000 µg/mL concentration with IC_50_ value of 32 µg/mL. Other extracts of *F. ammoniacum* also showed a concentration dependent response against the selected enzymes. 

### 3.5. In Vitro DPPH and ABTS Free Radicals Scavenging Potential of Extracts 

DPPH and ABTS free radical inhibitory potential of Fa.Met, Fa.Hex, Fa.Chf, Fa.EtAc, Fa.Bn and Fa.Aq are presented in Figure 2C,D. The results revealed that highest percent free radical scavenging potential was exhibited by Fa.EtAc fraction against DPPH and ABTS with lowest IC_50_ values of 100 and 120 µg/mL, respectively (Appendix A). Ascorbic acid was used as a positive control that caused 91.32 ± 0.34 and 95.31 ± 0.75 % inhibition at 1000 µg/mL against DPPH and ABTS with IC_50_ value 30 and 45 µg/mL.

### 3.6. Nootropic Effect of the Extracts in Y-Maze Test

Y-maze test results are presented in Figure 3. Results indicated that no significant (*p* > 0.05) reduction were observed in the total numbers of arm entries in both scopolamine and treatment groups (Fa.EtAc and DZP) as compared to normal saline group (Normal control) (Figure 3A). Same returns percentages were recorded significantly (*p* < 0.01) high in scopolamine-treated group as compared to normal saline group. Fa.EtAc (100 and 200 mg/kg body weight), and DZP (2 mg/kg) exhibited significant (*p* < 0.01) reduction in the percentage of returns to the same arm as compared to DW + Scop-treated group (Figure 3B). Similarly, Fa.EtAc (100 and 200 mg/kg body weight), and standard drug DZP (2 mg/kg body weight) significantly reverse the effect of scopolamine and have displayed a significant (*p* ˂ 0.05; *p* < 0.01) rise in the percentage of alternate arm returns comparable to scopolamine. Although, DW + Scop-treated group significantly (*p* ˂ 0.01) decreased the number of returns to alternate arm at the percentage of 26.2% as compared to normal saline group (65%) (Figure 3C).

The results of spontaneous alternation performance shows that there was a significant difference observed among all the treatments groups. Scopolamine significantly reduced (*p* ˂ 0.01) the mice spontaneous alternation percentage compared to normal saline group. Fa.EtAc at dose of 100 and 200 mg/kg body weight significantly reversed the effect of scopolamine and increased the spontaneous alternation performance percentage (*p* ˂ 0.05; *p* < 0.01) when compared to scopolamine-treated group. Standard drug Donepezil (2 mg/kg body weight) also significantly (*p* < 0.01) reversed the effects of scopolamine with a percentage of 60.4 (Figure 3D).

### 3.7. Nootropic Effect of Extracts in Novel Object Recognition Test

The NORT was used to assess the recognition memory of mice after a single injection of scopolamine. The results obtained with the NORT are shown in Figure 4 and Appendix A. There were no significant differences in the time spent in exploring the two identical objects between Fa.EtAc- and DZP-treated groups and scopolamine-treated group (Figure 4A). However, in the test phase Fa.EtAc (100 and 200 mg/kg), and donepezil (2 mg/kg) groups spent more time with novel object as compared to scopolamine-treated group (*p* < 0.05, *p* < 0.01). The administration of scopolamine before the retention phase of the test resulted in a decrease of the exploration time of the novel object in comparison with the familiar object (*p* < 0.01) (Figure 4B). The discrimination index (DI) which is 0.46 + 0.03 in DW + Sal (Distilled water plus normal saline) group was significantly reduced (*p* < 0.001) to a value of 0.07 + 0.05 in the scopolamine-treated group. The DI was significantly high for Fa.EtAc (50, 100 and 200 mg/kg) and donepezil (*p* < 0.01; *p* < 0.001) groups in comparison to scopolamine group (Figure 4C). The discrimination index of the standard drug donepezil group was 0.50 + 0.02. All groups showed %DI above 50% while scopolamine group has shown significantly low value as compared to vehicle control. The *F. ammoniacum* extracts significantly increased the discrimination index from 0.16 in the scopolamine-treated group to 0.45 and 0.51 (*p* < 0.001) in mice treated with Fa.EtAc at the doses 100 and 200 mg/kg, respectively. The exploration time of the novel object was also significantly increased by *F. ammoniacum* extracts from 25.0 ± 1.6 in the scopolamine-treated group to 37.3 ± 3.5 and 38.5 ± 2.2 in mice treated with Fa.EtAc at the doses of 100 and 200 mg/kg, respectively (*p* < 0.05, *p* < 0.01). The Fa.EtAc (100 and 200 mg/kg) also significantly decreased the exploration time of the familiar object (*p* < 0.01). The donepezil (2 mg/kg) groups significantly increased the discrimination index and the exploration time of the novel object (34.2 ± 1.2) and decreased the exploration time of the familiar object (11.30 ± 1.5) (*p* < 0.001). Here also, the effect of *F. ammoniacum* extracts at the dose 200 mg/kg was greater than the effect of donepezil.

In the NOL task, the % exploration time for each object was recorded both in the familiar location and novel location. In the sample phase, no difference was observed in exploration time between objects A1 and B2 (Figure 5A). However, in the test phase the animal gives preference to explore the novel location object as compared to the object in familiar location (Figure 5B).

### 3.8. Effect of Extracts on Brain Cholinesterases (AChE and BChE) Activity in Y-Maze Test in Mice 

Ex vivo analysis of cholinesterase (AChE and BChE) inhibitory potentials was measured using Fa.EtAc extract to assess the impairment in cholinergic functions in mice model. Percent AChE activity of frontal cortex and hippocampus tissues of different animal groups in Y-Maze test are given in Figure 6A,B, while %BChE activity are presented in Figure 7A,B. %AChE activity in the frontal cortex and hippocampus of scopolamine-treated groups were 51.2 + 0.2 and 44.4 + 0.4, while %BChE activities were 49.0 + 0.4 and 60.4 + 0.2 which were significantly (*p* < 0.01) ** higher than that of the normal saline group. In Fa.EtAc + Scop (50mg/kg, p.o.), Fa.EtAc +Scop (100 mg/kg, p.o.), and Fa.EtAc + Scop (200 mg/kg, p.o.)-treated groups, and DZP + Scop (20 mg/kg; (i.p.) a standard drug-treated group, a significant decline (*p* < 0.01) ^##^; (*p* < 0.001) ^###^ in %AChE and %BChE activities were observed in frontal cortex and hippocampus tissues which were lower than that of the scopolamine-treated group. 

### 3.9. Effect of Extracts on Brain Cholinesterases (AChE and BChE) Activity in NORT in Mice 

Results of percent AChE and BChE activity in frontal cortex and hippocampus of different animal groups in NORT are given in Figure 6C,D and Figure 7C,D. Percent AChE and BChE activities in the cortex and hippocampus of scopolamine-treated groups were significantly (*p* < 0.001 ***) higher than that of the normal saline group. %AChE and BChE activities in the cortex and hippocampus of Fa.EtAc + Scop (100 mg/kg, p.o.) and Fa.EtAc + Scop (200 mg/kg, p.o.)-treated groups, and standard DZP + Scop (20 mg/kg; (i.p.)-treated groups were significantly (*p* < 0.01) ^##^; lower as compared to scopolamine-treated group. 

### 3.10. Effect of Extracts on Brain Antioxidant Activity in Y-Maze Test 

Percent ex vivo antiradicals activity in cortex and hippocampus of different animal groups in Y-Maze test are presented in Figure 8A,B. Scopolamine-treated group significantly (*p* < 0.01 **) reduced the %DPPH free radicals scavenging activity/mg protein compared to the normal saline-treated group. Pre-treatment of mice with *F. ammoniacum* extract fractions like Fa.EtAc + Scop (50, 100, 200mg/kg, p.o.), and standard donepezil (20 mg/kg; (i.p.) significantly ((*p* < 0.05) ^#^; *p* < 0.01) ^##^ reversed the effect of scopolamine. 

Furthermore results regarding NORT of ex vivo %DPPH free radicals scavenging activity in cortex and hippocampus of different animals groups are presented in Figure 8C,D. %DPPH inhibition activity in the cortex and hippocampus of scopolamine-treated groups were significantly (**: *p* < 0.01) lower (14.6 + 0.2 and 19.4 + 0.5) as compared to normal saline group (44.6 + 0.3 and 58.0 + 0.4.). Fa.EtAc + Scop (50 mg/kg, p.o.), Fa.EtAc + Scop (100 mg/kg, p.o.), and Fa.EtAc + Scop (200 mg/kg, p.o.) and standard donepezil-treated groups showed significantly higher %anti-radicals activities of 24.6 + 0.1 ^#^, 31.6 + 0.5 ^##^, 38.2 + 0.4 ^##^, and 49.4 + 0.1 ^###^ in cortex and 33.6 + 0.5 ^#^, 51.4 + 0.5 ^##^, 56.8 + 0.2 ^##^, and 62.0 + 0.3 ^##^ in the hippocampus, respectively. 

## 4. Discussion

Different *Dorema* species are in use in the Middle East countries as folk remedies of asthma, bronchitis, diabetes and other infections [38]. Reported studies revealed that people of Iran are well-known to the resin-gum of *D. ammoniacum* from centuries ago and its collection was started there from nearly 4000 years ago. Being rich in ammoniacum, the medicinal gum-resin has been used for the treatment of various disorders, such as upper respiratory tract, gastrointestinal and central nervous system complications. Several chemical compounds (terpenes, coumarins and phenolic compounds) with a wide range of pharmacological activities including antioxidant, anti-microbial, cytotoxicity, anti-inflammatory, anti-diabetic, anticonvulsant, and hypolipidemic activities have been reported from this genus and are used in modern medicines [26,38]. Regardless of an extensive preclinical and clinical study to assess the safety and toxicity efficacy of such medicinal plants very few investigational studies are available. However, in recent times, there has been an increasing concern related to their safety and toxicity assessment [39]. Such toxicity studies will enhance the public reliabilities on the herbal medicinal plants which would increase their traditional and beneficial uses. Due to the extensive use *D. ammoniacum* in Persian traditional medicine, this study was designed to assess its nootropic effect in mice model. Before in vivo assessment, the chemical composition of *F. ammoniacum* (*D. ammoniacum*) methanolic extract and ethyl acetate fraction (active fraction) was determined through HPLC and in vitro antioxidant and anticholinesterase activities were performed along with determination of total phenolic and flavonoid contents. Presence of phenolic compounds such as sesquiterpene coumarins, phenols, flavonoids and phloroacetophenone glycosides have also been reported from *D. ammoniacum* and other Dorema species previously [38,40].

Phenolic compounds present in *F. ammoniacum* and other plants serve as antioxidants due to its hydroxyl groups, which have exhibited strong antioxidant potential and have been reported to cause cholinesterase inhibition as well as they are capable of reacting with active oxygen radicals such as hydroxyl radicals. Flavonoids are polyphenolic compounds found in fruits and vegetables, and are highly effective scavengers of most oxidizing molecules, including singlet oxygen, and various free radicals implicated in several diseases including AD. Flavonoids can provide protection to the brain cells by modulating intracellular signals and promoting cellular survival [41]. In the current study highest contents of total phenolic (88.23 ± 1.13) and flavonoids (85.93 ± 0.67) were detected in Fa.EtAc followed by Fa.Chf extract (TPC: 75.12 ± 1.58 and TFC: 77.83 ± 2.16). Phenolics and flavonoids are free radical scavengers responsible for the observed high antioxidant and anticholinesterase activities, thus having protective effects in many neuro degenerative diseases such AD. The inhibition of cholinesterase enzymes is considered as a promising strategy in the management of neurological and neurodegenerative disorders such as AD, where a deficit in cholinergic neurotransmission is often observed. Several pharmacological effects of the plant extract such as antioxidant, anticholinesterase, free radical scavenging, antidiabetic, and hypolipidemic activities regarding the essential oil and crude extract of roots and aerial parts of Dorema species have also been associated with the presence of phenolic and flavonoid compounds [23,28,29,42].

The cholinesterase inhibitory activity of several medicinal plants has been reported in the literature [43,44]. Additionally, antioxidants such as vitamin E and vitamin C low level are associated with incidences and prevalence of AD. AD patients administered with high doses of antioxidants were reported to have a slower rate of cognitive deterioration. Thus, the good antioxidant and anticholinesterase activities of the *F. ammoniacum* extract/fractions in this study suggest that these extract/fractions are good sources of phenolic and flavonoid compounds, with potential cholinesterase inhibitory and antioxidant properties that may find usefulness in the management of AD [43,44]. Furthermore, antioxidants and neuroprotective activities have been reported for the identified possible compounds in crude (chlorogenic acid, quercetin, mandelic acid, phloroglucinol, and hydroxy benzoic acid) and ethyl acetate extract (malic acid, chlorogenic acid, epigallocatechin gallate, quercetin, ellagic acid, rutin, pyrogallol), identified in other plant extracts. Among the identified phytochemicals; quercetin, chlorogenic acid, mandelic acid, epigallocatechin gallate, ellagic acid, and rutin has been reported as a potent antioxidant and neuroprotective agents [36,45,46,47,48]. In this study Fa.EtAc fraction showed free radical scavenging activity and strong inhibitions of cholinesterases (AChE and BChE), with low IC_50_ values. Reported studies also revealed that ethyl acetate extract *F. ammoniacum* has shown the highest antioxidant activity in both DPPH and FRAP assays [23]. Fa.EtAc fraction was found rich in phenolics and flavonoids that might thus be capable of reducing the risk of various degenerative diseases including AD, by prompting antioxidative activities. Previous studies also revealed that phenolics and flavonoids act as free radical scavengers of many oxidizing species and are inhibitors of cholinesterases (AChE and BChE) [49]. Further studies are needed to isolate and characterize the bioactive phytochemical compounds to be used as a promising candidate drug in management of AD. 

The key clinical symptom of AD is a progressive deterioration in learning and memory function due to deficiency of ACh levels in the brain. Ach is an essential neurotransmitter present in the synapses. Its low level can reduce cognitive functions. Cholinesterase inhibitors are employed to reduce ACh hydrolysis, thereby increasing the concentration of ACh at the synapses and presynaptic receptors that prevents the death of cholinergic neurons [50]. Hence, treatment with plant extract (Fa.EtAc) might prevent the breakdown of ACh and increases cholinergic transmission, leading to the amelioration of symptoms [3] like that of donepezil which is a standard inhibitor of AChE and BChE [5]. AChE and BChE appear to be simultaneously active in the synaptic hydrolysis of ACh, terminating its neurotransmitter action, and co-regulating levels of ACh which is considered to be the most essential neurotransmitter that controls the regulation of cognitive functions. Increased AChE activity cause the loss of cholinergic neurotransmissions leading to Dementia and Alzheimer’s disease. Inhibitions of AChE decrease the hydrolysis of ACh in the brain and increase cholinergic neurotransmissions which might be helpful in treating mild to moderate levels of Alzheimer’s disease. In order to search out effective inhibitors of AChE and BChE from natural sources particular attention is needed in adapting an effective strategy to treat AD and that is by enhancing the level of ACh in the synapses of neurons [50]. The selected plant extracts provide a considerable source of secondary metabolites like phenolic acids and flavonoids which plays a role as antioxidant and as cholinesterase inhibitors. These phytochemicals are less toxic due to its origin from medicinal plants as compared to synthetic drugs and if isolated would be safer alternatives of the drugs to treat the neurodegenerative disorders. Present study revealed that Fa.EtAc fraction showed an excellent inhibition of AChE and BChE that might be due to the presence of anticholinesterase compounds (quercetin, chlorogenic acid, epigallocatechin gallate, ellagic acid, and rutin). Other plant in form of extracts have also shown anticholinesterase potentials for example, *Ginkgo biloba* extract, which has been used as an efficient cholinesterase inhibitor, memory enhancer, and provide benefits for cognition and treatment of mild-to-moderate Alzheimer’s disease [51]. Ammoniacum gum has also been used to treat disorders like epilepsy and to reduce joint pain. The cytotoxic, and acetylcholinesterase inhibitory activities about them have already been reported [38].

Inhibition of acetyl cholinesterase enzyme can restore cholinergic functions and allow more retention of acetylcholine in the brain, which is essential for enhancing cognitive functions, learning, and memory. The present study validates the useful effect of ethyl acetate fraction of *F. ammoniacum* on scopolamine-induced memory impairments. Scopolamine significantly decreased memory in rodent’s model, indicating impairment in learning and memory in the behavioural Y-maze test. Our results were in line with the previous reported studies of other authors [36,48]. Scopolamine is a muscarinic cholinergic receptor antagonist which significantly causes memory impairments in mice especially in the progressions of learning acquisition and short-term memory. Scopolamine induces oxidative stress and inflammatory responses in the brain tissue which can cause chronic activation of microglia that disrupt synaptic plasticity and finally neuronal death. The ROS directly damage the neuron by increasing intracellular Ca^2+^ level, while inflammatory cytokines are produced due to the activation of microglia that inhibit the production of BDNF and decrease the level of ACh [52]. 

First the extract was evaluated for in vitro antioxidant and cholinesterase inhibitory potentials then behavioural tests like Y-maze and NORT were performed in in vivo model of mice and after getting promising results the mentioned potentials were checked in the frontal cortex and hippocampus. High AChE and BChE activities were observed in the frontal cortex and hippocampus tissues of the diseased control group (measured from supernatant part of homogenized brain tissues at a standardized protein content of 5 mg/mL). In standard control group (Donepezil) and test control (*F. ammoniacum* extract) group a significant decline in both AChE and BChE activities (Figure 6; Figure 7), and DPPH free radical scavenging activity (Figure 8) in the cortex and hippocampus, respectively, were observed which were also in line with previous reported studies [36,52]. The pre-treatment with Fa.EtAc significantly increased the learning and memory in rodent’s model. Reported studies also revealed that some plant extract significantly reversed the memory loss in mice model justifying its neuroprotective potential [36,52]. In the current study of NORT, the increase in the percent DI and exploration time with novel object compared to familiar object in mice treated with Fa.EtAc suggested improvement in learning and memory which is in agreement with the reported study [53]. On the other hand, scopolamine significantly decreased the DI, representing impairment in learning and memory. This effect was overturned by *F. ammoniacum* ethyl acetate fraction, due to their memory enhancing activity and nootropic effect. Nootropic agents or cognitive enhancers have already been reported in traditional medicinal system to improve mental functions such as cognition, memory, or attention [53]. 

Behavioural study was further confirmed by biochemical assessment of brain cholinesterases (AChE and BChE) enzymes activity and DPPH free radicals scavenging activity in mice. *F. ammoniacum* has clearly demonstrated anticholinesterase and antiradical activity with strong neuroprotective and nootropic potential. *F. ammoniacum* inhibited brain cholinesterases (AChE and BChE) enzymes levels suggesting that its nootropic effects may strongly be due to the antioxidant action and, therefore, might play an effective role as antioxidant and free radical scavenger. Reported studies revealed that *F. ammoniacum* gum resin also shows AChE inhibition due to presence of bioactive compounds [23]. This study is further supported by the possible involvement of brain-derived neurotropic factor (BDNF; a neuronal growth factor) found in the brain, majorly found in the hippocampus, amygdala, cerebellum and cerebral cortex in both rodents and humans. BDNF activate enteric nervous system signalling pathways and synaptic communication which is released by the microglia mediated microglia-neuron signalling receptor. BDNF activates the neurotransmitter system and choline acetyltransferase (ChAT) enzymes which synthesize the neurotransmitter acetylcholine (ACh). Acetylcholine is implicated in cognitive functions and is a treatment target for many psychiatric and neurological disorders [54]. Inhibitions of cholinesterases (AChE and BChE) probably enhance the levels of neurotransmitter (ACh) that might increase BDNF level. The donepezil and Fa.EtAc extract enhanced the Ach level while scopolamine a muscarinic cholinergic receptor antagonist suppressed its level in the cortex and hippocampus due to blockade of cholinergic signalling pathways, therefore, it is suggested that enhancement of neurogenesis by activating ACh is involved in the activation of muscarinic acetylcholine receptors. Although currently available therapies for AD are cholinesterase inhibitors such as donepezil and rivastigmine, which only reduces the disease progression and provide symptomatic relief. Efforts are still being made to find out better alternative and better therapeutic options. The probable mechanistic overview of major factors involved in the cognitive and memory impairments are presented in Figure 9. Moreover, based on the phytochemical, behavioural, and biochemical results, we hypothesize that *F. ammoniacum* could possibly act directly as a free radical scavenger or regulator to inhibit cholinesterases (AChE and BChE), oxidative stress, and memory impairments induced by scopolamine. No memory impairments in the scopolamine-treated mice exposed to ethyl acetate fraction of *F. ammoniacum*, were observed suggesting that *F. ammoniacum* possess nootropic and neuroprotective activities.

## 5. Conclusions

In conclusion, the fraction Fa.EtAc having highest level of total phenolic/flavonoid contents resulted in good inhibition of AChE and BChE enzymes with low IC_50_ values that also exhibited highest % free radical scavenging potential against DPPH and ABTS could be used as alternative drug to treat oxidative stress and neurodegenerative diseases. This extract has also exhibited nootropic and memory enhancing effect in mice model. In in vivo studies, it has exhibited strong antioxidant and anticholinesterase potential and was capable of reversing scopolamine-induced learning and memory impairments through reduction of brain cholinesterases (AChE and BChE) enzymes level. These results confirm the common use of *F. ammoniacum* as neuroprotective and nootropic medicinal plant in traditional medicine. However, further work to isolate the novel and safe nootropic agents present in *F. ammoniacum* and to understand the mechanisms of action responsible for these effects is needed. The findings of current study demonstrate the role of Fa.EtAc extract that probably activate the enteric nervous system signalling, increased the level of ACh and enhance neuronal activity that might be involved in memory processing.

## Figures and Tables

**Figure 1 brainsci-11-00259-f001:**
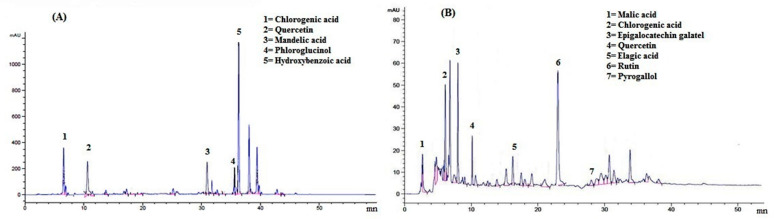
HPLC chromatogram of *Ferula ammoniacum* (D. Don) aerial parts extract/fraction. (**A**) crude methanolic extract (Fa.Met) and (**B**) ethyl acetate fraction (Fa.EtAc).

**Figure 2 brainsci-11-00259-f002:**
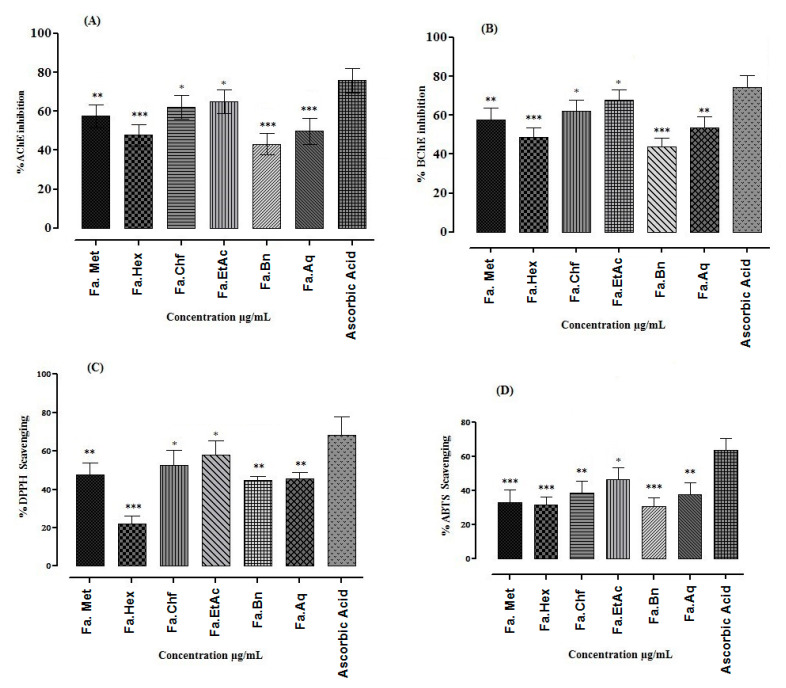
Percent anticholinesterase and antioxidant potential of extract/fractions of *Ferula ammoniacum* (D. Don) aerial parts. {(**A**) Percent inhibition potential of extract/fractions for AChE and (**B**) BChE; (**C**) DPPH and (**D**) ABTS free radical. The data are expressed as Mean ± SEM, (*n* = 3). One-way ANOVA followed by Dunnett’s post hoc multiple comparison test to determine the values of *p*. Values are significantly different as compare to positive control, * *p* < 0.05, ** *p* < 0.01, *** *p* < 0.001}.

**Figure 3 brainsci-11-00259-f003:**
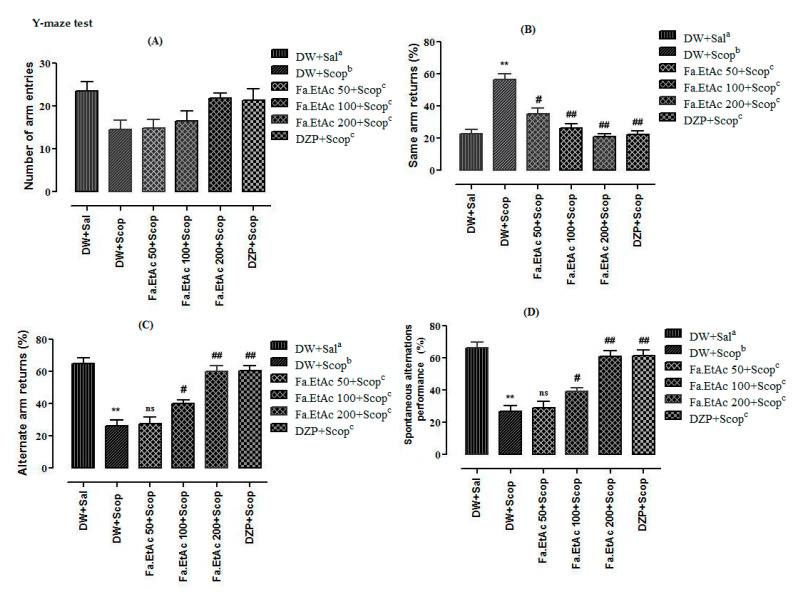
Effect of Fa.EtAc fraction of *Ferula ammoniacum* (D. Don) on mice in the behavioural Y-maze test. (**A**) Number of arm entries (**B**) Same arm returns (**C**) alternate arm returns (**D**) % Spontaneous alternation performance. The data are expressed as Mean ± SEM; each value corresponds to a mean of eight animals. One-way ANOVA followed by Dunnett’s post hoc multiple comparison test to determine the values of *p*. ** *p* < 0.01; comparison of DW + Sal^a^ (Normal control) vs. DW + Scop^b^ (Scopolamine treated), ^#^
*p* < 0.05 and ^##^
*p* < 0.01; comparison of (DW + Scop)^b^ vs. DZP + Scop^c^ (Donepezil treated)- and Fa.EtAc^c^ (50, 100 and 200 mg/kg)-treated groups), ns: values not significantly different in comparison to (DW + Scop)^b^-treated group using one-way ANOVA followed by Dunnett’s post hoc multiple comparison test.

**Figure 4 brainsci-11-00259-f004:**
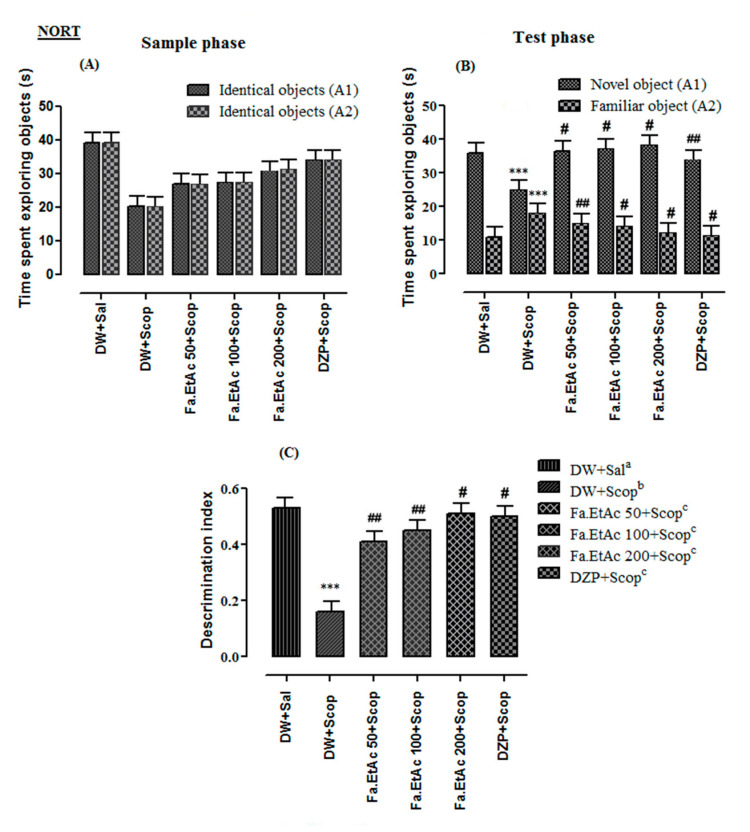
Effect of Fa.EtAc fraction of *Ferula ammoniacum* (D. Don) on mice in behavioural NORT (**A**) Time spent in the sample phase (**B**) Time spent in the test phase (**C**) % Discrimination index were recorded in Fa.EtAc (50, 100 and 200 mg/kg)-treated groups versus scopolamine (Scop. 1 mg/kg)-treated group for assessment of recognition memory in mice model in behavioural NORT. The data are expressed as Mean ± SEM; each value corresponds to a mean of eight animals. One-way ANOVA followed by Dunnett’s post hoc multiple comparison test to determine the values of *p*. *** *p* < 0.001; comparison of DW + Sal^a^ (Normal control) vs. DW + Scop^b^ (Scopolamine-treated group), ^#^
*p* < 0.05 and ^##^
*p* < 0.01; comparison of (DW + Scop)^b^ vs. DZP + Scop^c^ (Donepezil treated) and Fa.EtAc^c^ (50, 100 and 200 mg/kg)-treated groups using one-way ANOVA followed by Dunnett’s post hoc multiple comparison test.

**Figure 5 brainsci-11-00259-f005:**
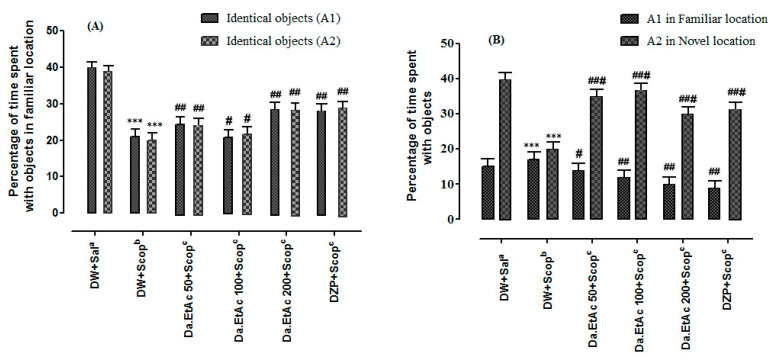
Effect of Fa.EtAc fraction of *Ferula ammoniacum* (D. Don) on mice in behavioural NOL task (**A**) Percent exploration time between the objects A1 and A2 in familiar location in the sample phase (**B**) Percent exploration time between the objects A1 and A2 in novel location in the test phase. The data are expressed as Mean ± SEM; each value corresponds to a mean of eight animals. One-way ANOVA followed by Dunnett’s post hoc multiple comparison test to determine the values of *p*. *** *p* < 0.001, ^#^
*p* < 0.05, and ^##^
*p* < 0.01 and ^###^
*p* < 0.001; comparison of A1 (Object in familiar location) vs. A2 (Object in novel location).

**Figure 6 brainsci-11-00259-f006:**
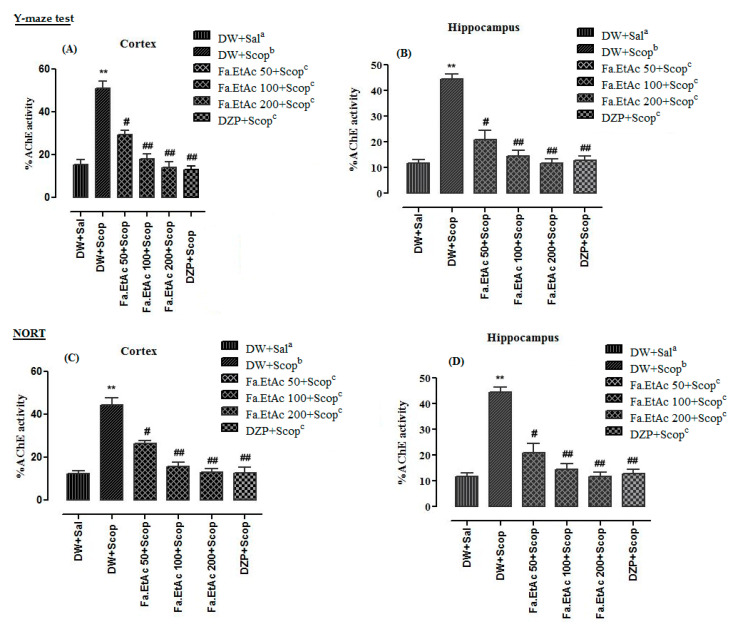
Ex vivo % AChE activity in the (**A**) frontal cortex and (**B**) hippocampus of different animal groups in Y-maze behavioural test and (**C**,**D**) NORT. The data are expressed as Mean ± SEM; each value corresponds to a mean of eight animals. One-way ANOVA followed by Dunnett’s post hoc multiple comparison test to determine the values of *p*. ** *p* < 0.01; comparison of DW + Sal^a^ (Normal control) vs. DW + Scop^b^ (Scopolamine treated), ^#^
*p* < 0.05 and ^##^
*p* < 0.01; comparison of (DW + Scop)^b^ vs. DZP + Scop^c^ (Donepezil treated) and Fa.EtAc^c^ (50, 100 and 200 mg/kg)-treated groups), ns: values not significantly different in comparison to (DW + Scop)^b^-treated group using one-way ANOVA followed by Dunnett’s post hoc multiple comparison test.

**Figure 7 brainsci-11-00259-f007:**
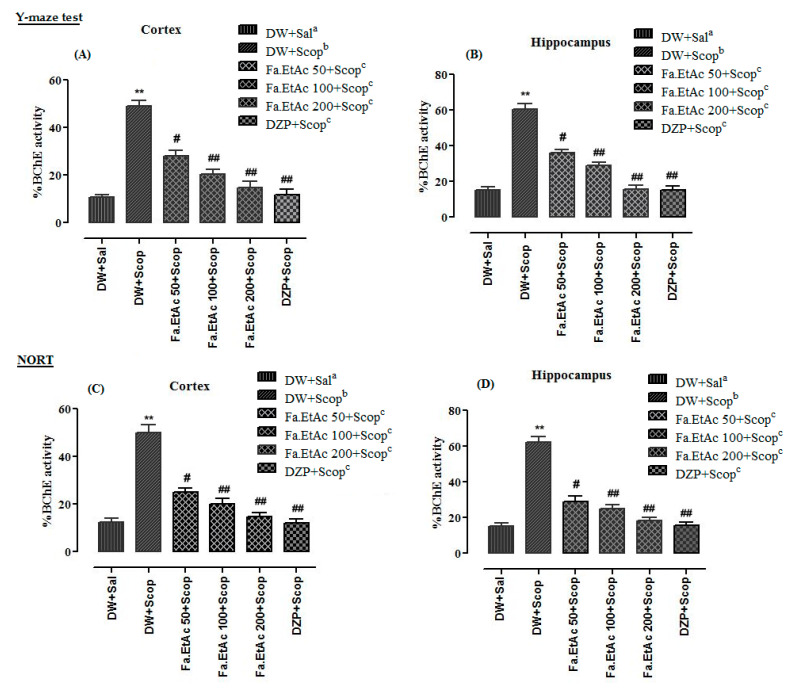
Ex vivo % BChE activity in the (**A**) frontal cortex and (**B**) hippocampus of different animal in Y-maze behavioural task and (**C**,**D**) NORT. The data are expressed as Mean ± SEM; each value corresponds to a mean of eight animals. One-way ANOVA followed by Dunnett’s post hoc multiple comparison test to determine the values of *p*. ** *p* < 0.01; comparison of DW + Sal^a^ (Normal control) vs. DW + Scop^b^ (Scopolamine treated), ^#^
*p* < 0.05 and ^##^
*p* < 0.01; comparison of (DW + Scop)^b^ vs. DZP + Scop^c^ (Donepezil treated) and Fa.EtAc^c^ (50, 100 and 200 mg/kg)-treated groups), ns: values not significantly different in comparison to (DW + Scop)^b^-treated group using one-way ANOVA followed by Dunnett’s post hoc multiple comparison test.

**Figure 8 brainsci-11-00259-f008:**
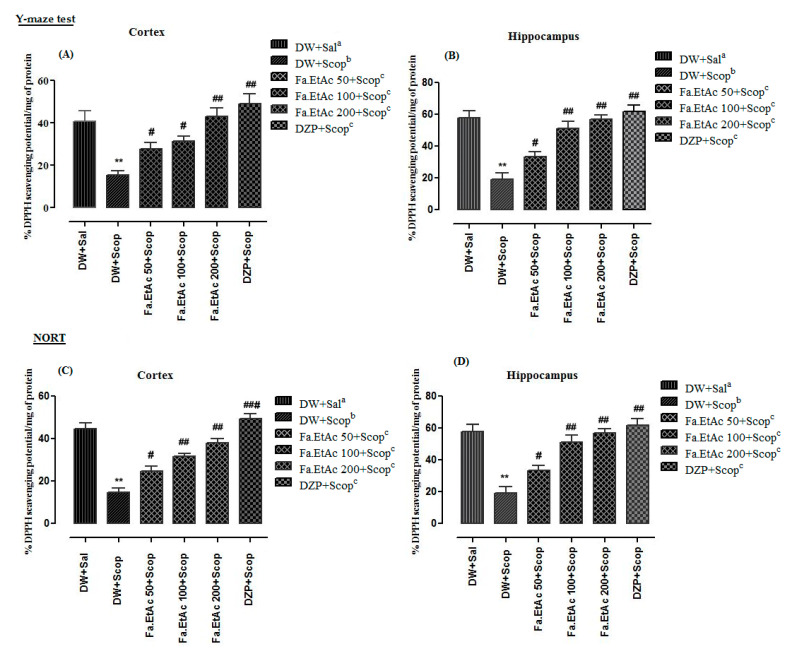
Ex vivo %DPPH free radical scavenging effects in (**A**) frontal cortex and (**B**) hippocampus of different animal groups in Y-maze behavioural task and (**C**,**D**) NORT. The data are expressed as Mean ± SEM; each value corresponds to a mean of eight animals. One-way ANOVA followed by Dunnett’s post hoc multiple comparison test to determine the values of *p*. ** *p* < 0.01; comparison of DW + Sal^a^ (Normal control) vs. DW + Scop^b^ (Scopolamine treated), ^#^
*p* < 0.05, ^##^
*p* < 0.01, and ^###^
*p* < 0.001; comparison of (DW + Scop)^b^ vs. DZP + Scop^c^ (Donepezil treated) and Fa.EtAc^c^ (50, 100 and 200 mg/kg)-treated groups), ns: values not significantly different in comparison to (DW + Scop)^b^-treated group using one-way ANOVA followed by Dunnett’s post hoc multiple comparison test.

**Figure 9 brainsci-11-00259-f009:**
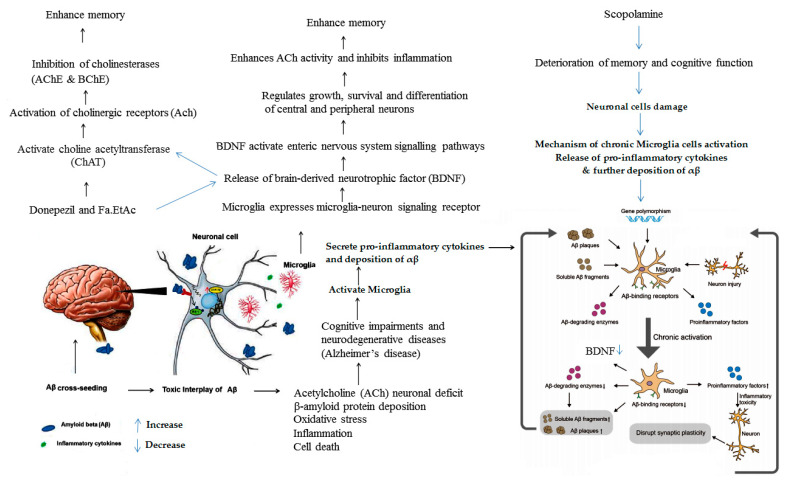
Mechanistic overview of Fa.EtAc as nootropic on the responsible factors involved in the aetiology of neurodegenerative diseases.

**Table 1 brainsci-11-00259-t001:** Experimental design showing dosing details.

Group	Group Category	Treatment Given	Route
I	DW + Sal	Sal (8 mL/kg)	p.o.
II	DW + Scop	Scop (1 mg/kg)	i.p.
III	DZP + Scop	DZP (2 mg/kg) + Scope (1 mg/kg)	p.o., i.p.
IV	Fa.EtAc 50 + Scop	Fa.EtAc (50 mg/kg) + Scop (1 mg/kg)	p.o.
V	Fa.EtAc 100 + Scop	Fa.EtAc (100 mg/kg) + Scop (1 mg/kg)	p.o
VI	Fa.EtAc 200 + Scop	Fa.EtAc (200 mg/kg) + Scop (1 mg/kg)	p.o.

Fa.EtAc, *Ferula ammoniacum* ethyl acetate fraction; DW, Distilled water; Sal, Normal Saline; Scop, scopolamine; DZP, Donepezil; p.o., Per oral; i.p., Intraperitoneal.

**Table 2 brainsci-11-00259-t002:** Possible compounds identified in *Ferula ammoniacum* (D. Don) aerial parts extracts.

Extract	Peak No	Retention Time (min)	Detected Phenolic Compounds	Sample Peak Area	Standard Peak Area	Concentration (µg/mL)
Fa.Met	1	6.5	Chlorogenic acid	132.221	12.9	9.20
	2	10.5	Quercetin	64.97	90.9	7.14
	3	30.5	Mandelic acid	110.82	72.0	15.39
	4	35.5	Phloroglucinol	415.64	25.02	14.95
	5	36.3	Hydroxy benzoic acid	4190.44	40.19	93.83
Fa.EtAc	1	3.1	Malic acid	50.88	40.32	12.63
	2	6.5	Chlorogenic acid	180.83	12.9	140.17
	3	8.0	Epigallocatechin gallate	90.70	7261.47	69.52
	4	10.5	Quercetin	27.82	90.9	3.05
	5	16.6	Ellagic acid	24.46	319.24	0.76
	6	22.7	Rutin	107.12	2241.2	47.82
	7	28.1	Pyrogallol	9.1	1.014	91.0

**Table 3 brainsci-11-00259-t003:** Total Phenolic Content and Flavonoids Content in the extracts/fractions of *Ferula ammoniacum* (D. Don).

S. No	Extract/Fractions	TPC (mg GAE/g)	TFC (mg QE/g)
1	Fa.Met	68.25 ± 1.14	66.97 ± 0.99
2	Fa.Hex	65.55 ± 0.84	68.01 ± 1.17
3	Fa.Chf	75.12 ± 1.58	77.83 ± 2.16
4	Fa.EtAc	88.23 ± 1.13	85.93 ± 0.67
5	Fa.Bn	55.18 ± 0.68	45.51 ± 1.99
6	Fa.Aq	51.75 ± 1.87	59.79 ± 1.39

Fa, *Ferula ammoniacum*; TPC, Total Phenolic Content; TFC, Total Flavonoid Content; Fa.Cr, Crude methanolic extract; Fa.Hex, *n*-hexane fraction; Fa.Chf, Chloroform fraction; Fa.EtAc, Ethyl acetate fraction; Fa.Bn, *n*-Butanol; Fa.Aq, Aqueous fraction; GAE, Gallic acid equivalent; QE, Quercetin equivalent.

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
