# Peer review of "Phytochemical Analysis, In Vitro Anticholinesterase, Antioxidant Activity and In Vivo Nootropic Effect of Ferula ammoniacum (Dorema ammoniacum) D. Don. in Scopolamine-Induced Memory Impairment in Mice"

_brainsci, 2021, doi:10.3390/brainsci11020259_

Round 1

Reviewer 1 Report

The study by Nazir et al. investigates the protective role of Ferula ammoniacum (D. Don) in scopolamine-induced memory impairment in mice. The authors measured the biochemical properties of D. Don and subsequently conducted several behavioral analyses in scopolamine-treated mice that have already been pretreated with the Fa. EtAC extract. Overall, the manuscript was written in good English and easy to follow. Some of my minor comments are:

  1. Please include your rationale for investigating the activities of AChE, BChE and free radical scavenging in cortex and hippocampus. No rationale and discussion of results in these two different brain regions were provided.
  2. Figure legend in Figure 1. (D. Don)
  3. Figure 2: The x-axis is not correctly labeled
  4. The authors used donezepil (DZP) as a positive control. Please provide more rationale and published evidence to validate the use of DZP as a positive control along with supporting data from this study.
  5. Figure 4: Please include the unit for y-axis.
  6. Figure 5:
    1. Please define the superscripted a,b,c
    2. Please define the symbols shown on the bar (***,#, ##,###)
  7. Please correct the spelling for “Y-maze” in this manuscript.

Author Response

Reviewer 1:

Comments and Suggestions for Authors

The study by Nazir et al. investigates the protective role of Ferula ammoniacum (D. Don) in scopolamine-induced memory impairment in mice. The authors measured the biochemical properties of D. Don and subsequently conducted several behavioral analyses in scopolamine-treated mice that have already been pretreated with the Fa. EtAC extract. Overall, the manuscript was written in good English and easy to follow. Some of my minor comments are:

Dear Editor/ reviewer,

Thank you very much for kind review and comments concerning our manuscript. Thank you so much worthy editors/reviewers for appreciation the manuscript data, experimental method used/plan of study, and about the findings of study. We appreciate the hard work of reviewers as they fairly pointed out errors and mistakes in our manuscript. We have tried to revise the manuscript in line with comments of the reviewers. Corrections made have been highlighted as Blue.

Please find below the point by point responses to the reviewer’s comments and suggestions.

  1. Please include your rationale for investigating the activities of AChE, BChE and free radical scavenging in cortex and hippocampus. No rationale and discussion of results in these two different brain regions were provided.
  • Answer: Respected reviewer the rationale related the activities of AChE, BChE and free radical scavenging potential in cortex and hippocampus has been included in the introduction and discussion section. Some of the detail is also given below:
  • Respected reviewer, antioxidant potential and cholinesterases (AChE and BChE) enzymes inhibition are the possible mechanism for the treatment of Alzheimer, s disease (AD). As there is a strong correlation between the reactive oxygen species production and synthesis of cholinesterases (AChE and BChE) enzymes produced in oxidative stress related neurodegenerative diseases like AD. In patients with AD, intense AChE and BChE activity and free radicals have been found in the area of amyloid-beta (αβ) plaques. Amyloids-beta (αβ) and cholinesterases activity may act as a physiological modulator of cholinergic function and induce neurotoxicity, and inhibit the synthesis and release of ACh, resulting in cholinergic hypofunction, reduced neural efficiency, and cognitive impairment. These effects seem to appear in the cortex and hippocampus regions but not in other brain areas [1].
  • Therefore, keeping in view the pathological conditions involved in AD progression, we have planed the in vitro analysis of antioxidant and cholinesterase inhibitory potentials using ammoniacum extract/fractions, our next target was to check if our test sample also inhibits AChE and BChE enzymes in frontal cortex and hippocampus in in vivo model of mice using behavioural tests like Y-maze and NORT. In the current results high AChE and BChE activities were observed in the frontal cortex and hippocampus tissues of disease control group which have shown impairments in brain tissues. In standard control group (Donepezil) and test control (F. ammoniacum extract) group a significant decline in the both AChE and BChE activities (Figure 6 & 7), and DPPH free radical scavenging activity (Figure 8) in the cortex and hippocampus respectively were observed which have shown the protective effects of F. ammoniacum plant extract.
  • This protective effect might be due to phenolic compounds present in ammoniacum plant serve as antioxidants due to its hydroxyl groups, which are responsible for their scavenging action. Thus, they are capable of reacting with active oxygen radicals such as hydroxyl radicals. Flavonoids are polyphenolic compounds found in fruits and vegetables, and are responsible to be highly effective scavengers of most oxidizing molecules, including singlet oxygen, and various free radicals generated in AD [2]. Phenolics and flavonoids are free radical terminators, thus having protective effects against many neuro degenerative diseases such AD. The inhibition of cholinesterase enzymes is considered promising in the management of neurological and neurodegenerative disorders such as AD, where a deficit in cholinergic neurotransmission is often observed.
  1. Figure legend in Figure 1. (D. Don)
  • Answer: Corrected accordingly.
  1. Figure 2: The x-axis is not correctly labelled
  • Answer: Worthy reviewer the x-axis has been corrected accordingly.
  1. The authors used donepezil (DZP) as a positive control. Please provide more rationale and published evidence to validate the use of DZP as a positive control along with supporting data from this study.
  • Answer: Cholinesterase inhibitors like Donepezil are approved by the Food and Drug Administration for the symptomatic treatment of AD. These drugs improve cognitive and neuropsychiatric symptoms. Researchers are also in search of medicinal plants based cholinesterase inhibitors which significantly recovers cognitive decline induced by streptozotocin. Moreover, synthetic antioxidants have also been utilized but they exhibited adverse side effects including liver damage and carcinogenesis. Natural sources, especially plants, provide a diverse and largely unexploited reservoir of bioactive phytochemicals for drug discovery and for the development of new cholinesterase inhibitors and antioxidants. Previous studies have already highlighted the potential beneficial effects of plants as vital sources for cholinesterase inhibitors [3].
  • Donepezil is a second-generation cholinesterases inhibitor with a high selectivity in the CNS can reversibly inhibit cholinesterase (AChE and BChE) enzymes activity and reduce ACh degradation and increase neurotransmitter concentration in the synaptic cleft that prolongs its duration of action, thus elevating central cholinergic activity to improve cognitive function. AChE and BChE appear to be simultaneously active in the synaptic hydrolysis of ACh, terminating its neurotransmitter action, and co-regulating levels of Ach. Acetylcholine is a key neurotransmitter in the nervous system that interacts with receptors associated with processes of learning and memory. The use of donepezil seems to have a positive effect on the functioning and enhancements in cognition. Hence, treatment with donepezil and plant extract might prevents the breakdown of ACh and increases cholinergic transmission, leading to the amelioration of symptoms [4].
  1. Figure 4: Please include the unit for y-axis.
  • Answer: Respected reviewer the units on y-axis of Figure A and B has already been given in which ‘’s’’ represent the seconds (s).
  • Worthy reviewer while in Figure C the Discrimination index (DI) has no unit as it is a ratio and has been calculated in percentage which is given in detail in the supplementary file Table S3.
  • Parameters including the time (seconds) spend in exploring familiar (F) object, time (in seconds) spend on exploring the novel (N) object, and total time (in seconds) spend on exploring both objects (N+F) were measured separately. The DI is calculated by the following equation already given in the method section.   

·       (7)

  1. Figure 5:
  1. Please define the superscripted a,b,c
  • Answer: Respected reviewer the superscript ‘‘a’’ represented the normal control group (DW+Sala) animals, superscript ‘‘b’’ has shown the scopolamine treated (DW+Scopb) group animals, while superscript ‘‘c’’ has shown the positive control Donepezil treated (DW+Scop)b group and test sample treated(Fa.EtAcc (50, 100 and 200 mg/kg) groups animals.
    1. Please define the symbols shown on the bar (***,#, ##,###)
  • Answer: Respected reviewer the symbols you have mentioned has shown the statistical analysis. One way ANOVA followed by Dunnett's post hoc multiple comparison test was used to determine the values of P. Signs on the bar like *p< 0.05, **p<0.01, and ***p<0.001 were used when comparison were carried out between the normal control (DW+Sala) vs Scopolamine (DW+Scopb) treated groups animals. While signs like #p < 0.05, ##p < 0.01,  and ###p < 0.001 were used when comparison were done between the Scopolamine treated (DW+Scop)b groups vs treatment groups like Donepezil treated (DZP+Scopc) and Fa.EtAcc (50, 100 and 200 mg/kg)  treated groups.
  1. Please correct the spelling for “Y-maze” in this manuscript.
  • Answer: Worthy reviewer Y-maze is corrected accordingly.

References:

  1. Stanciu, G.D.; Luca, A.; Rusu, R.N.; Bild, V.; Beschea Chiriac, S.I.; Solcan, C.; Bild, W.; Ababei, D.C. ''Alzheimer’s Disease Pharmacotherapy in Relation to Cholinergic System Involvement'' 2020, 10(1), 40.
  2. Lopa, S.S.; Al-Amin, M.Y.; Hasan, M.K.; Ahammed, M.S.; Islam, K.M.M.; Alam, A.H.M.K.; Tanaka, T.; Sadik, G. ''Phytochemical Analysis and Cholinesterase Inhibitory and Antioxidant Activities of Enhydra fluctuans Relevant in the Management of Alzheimer’s Disease''  Int. J. Food Sci. 2021, 2021, 1-8.
  3. Okello, E.J., Mather, ''Comparative Kinetics of Acetyl- and Butyryl-Cholinesterase Inhibition by Green Tea Catechins|Relevance to the Symptomatic Treatment of Alzheimer’s Disease''  Nutrients. 2020, 12(4), 1090.
  4. Kim, G.W.; Kim, B.C.; Park, K.S.; Jeong, G.W. ''A pilot study of brain morphometry following donepezil treatment in mild cognitive impairment: volume changes of cortical/subcortical regions and hippocampal subfields’’ Scientific 2020, 10(1), 10912.

Reviewer 2 Report

dear authors I have read your work with interest. The experimental design
is rigorous and so the experiments that support your conclusions.
I think you should implement and improve the introduction and include studies
on eucriotic cell lines so you do in discussion. The manuscript so will more
integral.

Author Response

Reviewer 2:

Comments and Suggestions for Authors

Dear authors I have read your work with interest. The experimental design is rigorous and so the experiments that support your conclusions.

Dear Editor/ reviewer,

Thank you very much for kind review and comments concerning our manuscript. Thank you so much worthy editors/reviewers for appreciation the manuscript data, experimental method used/plan of study, and about the conclusion of the study. We have tried to revise the manuscript in line with comments of the reviewers. Corrections made have been highlighted as Blue.

Please find below the point by point responses to the reviewer’s comments and suggestions.

I think you should implement and improve the introduction and include studies on eucriotic cell lines so you do in discussion. The manuscript so will more integral.

  • Answer: Worthy reviewer the studies related cell lines has been included in the introduction section. Some detail is also give below:
  • Increasing evidence associates neurodegenerative diseases like AD with some pathological conditions, such as impaired mitochondrial function, amyloids–beta (αβ) deposit, neuroinflammation, cholinergic deficit, and oxidative stress. In patients with AD, intense AChE and BChE activity and free radicals have been found in the area of amyloid plaques and neurofibrillary tangles. Amyloids-beta (αβ) and cholinesterases activity may act as a physiological modulator of cholinergic function and induce neurotoxicity, and inhibit the synthesis and release of ACh, resulting in cholinergic hypofunction, reduced neural efficiency, and cognitive impairment [1, 2]. These effects seem to appear in the cortex and hippocampus regions but not in other brain areas. αβ activates microglial cells to act as pro-inflammatory cells that secrete pro-inflammatory cytokines, which induce further αβ deposition. Microglia cells are the resident immune cells of the central nervous system which express surface receptors that activate or amplify the innate immune response. During cellular damage microglial cells respond quickly by inducing a protective immune response, which result in the up-regulation of inflammatory molecules as well as neurotrophic factors. However, in case of chronic inflammation the prolonged activation of microglia produce a wide array of neurotoxic products, proinflammatory cytokines such as interleukin-1 (IL-1β), interleukin-6 (IL-6), tumor necrosis factor alpha (TNFα), and cholinesterases enzymes (AChE and BChE) into the extracellular space [2, 3]. Animal studies have shown that increased levels of protein (IL-1β, TNFα and IL-6) in the hippocampus and cerebral cortex are primarily from microglia. It has been proposed that the increase microglial cells activation in brain may be one of the early events that leads to oxidative damage and considered the most abundant source of free radicals in the brain such as superoxide and nitric oxide. Micro-glial cell derived radicals, as well as their reaction products such as hydrogen peroxide and peroxynitrite have been shown to be involved in oxidative damage and neuronal cell death in neurological diseases such as AD. Studies have shown that microglia cells play a role in supporting cognitive processes and homeostasis in the healthy adult brain and the absence of microglia results in cognitive and learning deficits in rodents during development. It also should be noted that microglial cells have efficient antioxidative defense mechanisms but when the production of ROS is prolonged, the endogenous reserves of antioxidants become exhausted and result in cellular damage [1, 4].

References:

  1. Augusto-Oliveira, M.; Arrifano, G.P.; Lopes-Araújo, A.; Santos-Sacramento, L.; Takeda, P.Y.; Anthony, D.C.; Malva, J.O.; Crespo-Lopez, M.E. ''What Do Microglia Really Do in Healthy Adult Brain?''  Cells. 2019, 8(10),  1293.
  2. Lane, R.M.; Potkin, S.G.; Enz, A. ''Targeting acetylcholinesterase and butyrylcholinesterase in dementia'' International Journal of Neuropsychopharmacology. 2006, 9(1), 101-124.
  3. Schmitz, T.W.; Soreq, H.; Poirier, J.; Spreng, R.N. ''Longitudinal Basal Forebrain Degeneration Interacts with TREM2/C3 Biomarkers of Inflammation in Presymptomatic Alzheimer's Disease'' The Journal of Neuroscience. 2020,  40(9),  1931.
  4. Fleiss, B.; Van Steenwinckel, J.; Bokobza, C.; Shearer, I. K.; Ross-Munro, E.; Gressens, P. ''Microglia-Mediated Neurodegeneration in Perinatal Brain Injuries'' Biomolecules. 2021. 11(1),  99.